# Effect of Pumpkin Cultivar on the Selected Quality Parameters of Functional Non-Dairy Frozen Desserts

Aleksandra Szydłowska [1,*] , Dorota Zielińska [1,*] and Danuta Kołożyn-Krajewska [1,2]

1   Institute of Human Nutrition Sciences, Warsaw University of Life Sciences (WULS), Nowoursynowska St. 159C, 02-776 Warszawa, Poland
2   Chair of Dietetics and Food Science, Jan Dlugosz University in Czestochowa, 42-200 Częstochowa, Poland
*   Correspondence: aleksandra_szydlowska@sggw.edu.pl (A.S.); dorota_zielinska@sggw.edu.pl (D.Z.)

**Abstract:** This study was conducted to investigate the influence of pumpkin cultivar as a fermented semi-product on the selected quality parameters of functional non-dairy frozen desserts, which were prepared using the potentially probiotic strain cultures *L. rhamnosus* Lock 0900 and *L. casei* O14. Microbiological status, pH, sugar and carotenoids content, antioxidant activity, color and sensory quality have been checked. Regardless of the pumpkin cultivar used, the lactic acid fermentation of pulps with selected bacterial strains decreased reducing sugar and total sugars content. Improvement in the antioxidant activity was also observed. The impact of the individual sensory attributes on the overall quality was determined. The overall quality of the investigated pumpkin sorbets was positively driven by the sweet taste, pumpkin flavor, smoothness of texture, and negatively driven by acidic flavor, pungent taste, and bitter taste, verified by PCA method. All of the final products achieved high survival of probiotics (higher than 8.4 log CFU/g) and revealed a good sensory quality (overall quality higher than 8 c.u). The treatments with the cultivar "*Melon Yellow*" of the *Cucurbita maxima* species, were characterized by significantly higher ($p < 0.05$) carotenoids content, total sugars and reducing sugars content and antioxidant activity, measured by two methods. It can be concluded that the pumpkin cultivar and strain culture used for the fermentation affect the count of potentially probiotic bacteria in the final products, the composition of bioactive compounds, antioxidant activity and sensory quality of the functional pumpkin frozen desserts.

**Keywords:** pumpkin cultivar; *Lactobacillus*; probiotic; frozen desserts; functional food; antioxidant activity; sensory properties

## 1. Introduction

The major changes in relation to an increased "nutritional awareness" of the consumer and progress in food technology contribute to the dynamic growth of the functional food market. The global plant-based food market is expected to grow at a cumulative annual growth rate (CAGR) of 11.9% from 2020 to 2027 to reach 74.2 billion USD by 2027 [1]. The development of novel functional plant-based food products is becoming a challenge for the food industry, taking into account the evolving vegan, flexitarian and vegetarian consumer diets and also the diets of those with conditions such as lactose intolerance. The growth of this food segment has a large market potential [1].

Probiotic products are examples of functional foods. The probiotic food market is dominated by dairy products. However, it also includes non-dairy products (pickled vegetables, juices, ice cream, desserts), as well as fermented soybean products such as natto, tempeh, and miso or meat products.

Probiotic microorganisms are defined as "live microorganisms that, when administered in adequate amounts, confer a health benefit on the host" [2]. The application of fermentation technology to plant-origin food matrix processing and the development of novel probiotic fermented products increase the added value of fruit and vegetables. This

solution also combines probiotics and their active metabolites with prebiotics (dietary fiber, etc.), thereby promoting intestinal health as well as preventing and relieving chronic diseases. It also indicates an important development direction for the probiotics industry in the future [3]. The pumpkins (*Cucurbita* sp.) are known for their high concentrations of carotenoids such as α- and β-carotene, α- and β- cryptoxanthin, lutein, antheraxanthin and a few other xanthophylls. They are also a rich source of mineral salts and vitamins for human consumption. These vegetables do not accumulate contaminants, e.g., from the soil, and are characterized by a very low, trace level of heavy metals. The pumpkin flesh is eaten raw, and also provides a valuable component of many dishes and desserts in gastronomy technology and it is used as an addition in fruit products for industrial production [4–6].

It is known that pumpkin beverages and sorbets are suitable food matrices for probiotics [7–9]. According to Paredes-Toledo (2021) [10], pumpkin seeds are also a good candidate for probiotic products. However, the possibility of using a pumpkin by-product to produce novel supplements containing *L. casei* has also been explored [11]. High resistance of the oligosaccharides from pumpkin pulp towards hydrolysis in human gastric juice can stimulate the lactobacilli growth [12]. For nutritional reasons, in sorbet technology, a vegetable was used as a raw material. Vegetables in the diet are a valuable source of antioxidative vitamins and have basic properties and low-calorie properties. Recently, some authors [13,14] also reported that a recommended healthy consumption of antioxidant nutrients to guarantee good "strengthening of defenses" may be necessary to address aspects of the world health emergency caused by the spread of the SARS-CoV-2 virus. The concept of a probiotic plant-based frozen dessert is also a part of the "clean label" trend, which means the product is positioned to can be "organic", "natural" (i.e., following natural production method) and/or "free from" artificial ingredients/additives [15].

The previous studies have explored the possibility of using pumpkin flesh to produce probiotic products and to examine their quality. The technology of probiotic pumpkin sorbet production has helped develop optimal conditions of fermentation to obtain the highest survivability of probiotic cultures, as well as the best sensorial attributes [9,16,17]. However, in plant-based food technology, a new trend has recently been noted in the development of novel, potentially functional products in the aspect of the plant cultivar used [18–20]. Therefore, the aim of this study was to investigate the effect of using two pumpkin cultivars on the selected quality parameters of functional frozen desserts. The hypothesis of the work is that pumpkin cultivar has a significant impact on the microbiological, physico-chemical and sensory quality of functional sorbets.

## 2. Materials and Methods

### 2.1. Pumpkin Varieties

The following two cultivars of pumpkin [21] were used to produce frozen desserts:

- "*Melon Yellow*" (belonging to the *Cucurbita maxima* species), is one of the oldest varieties. Its plants have shelling shoots up to 10 m long. It has large-scale fruit weighing between 15 and 30 kg, mostly spherical, ribbed, soft orange peel, only sometimes the fruit may be slightly flattened. The flesh of the fruit is compact, juicy, yellowish orange. This variety has an average dry matter, protein and vitamin content. Their content also depends to a large extent on the way in which the crop is grown. The quantity of these ingredients is higher in organic cultivation than conventional cultivation and, for nitrates, their concentration is smaller in organic farming,

- "*Miranda*" (belonging to the *Cucurbita moschata* species), is a variety of oil-free pumpkin seeds with no seed coat. The 3 to 4 spherical flattened fruits are formed on one plant of 3 to 4 kg. The fruit is light green, marble. As the skin grows, it color turns orange. Inside the fruit, apart from the tasty and thin flesh, is a lot of olive green, non-hulling seeds. They contain 25 to 50% fats and produce edible oil.

Both species were certified as organic foods and acquired from the Organic Farming Cooperative "*Dolina Mogilnicy*" (Wołkowo, Poland). Pumpkins were harvested between Oc-

tober and November (2017) when they reached maturity. The raw material was transported to the laboratory and properly processed.

While other ingredients, such as still water (Żywiec Zdrój, Poland), Gelatin (Gellwe, Poland), Sucrose (Diamant, Gostyń, Poland) were purchased in the local market in Warsaw, Poland.

### 2.2. Probiotic Bacterial Strains and Growth Conditions

Two bacterial strains have been applied:

- The probiotic, human-origin bacterial strain *Lacticaseibacillus rhamnosus* Lock 0900 (former *Lactobacillus rhamnosus* Lock0900; patent number 209988) was obtained from a pure culture maintained at the Laboratory of Microbiology, Łódź University of Technology, Poland [22]
- The potentially probiotic strain isolated from traditionally fermented cucumber pickles—*Lacticaseibacillus casei* O14 (former *Lactobacillus casei* O14) (GenBank accession KM 186154), described previously by Zielińska et al. (2015, 2019 and 2021) [23–25].

The bacterial strains were stored at $-80\ ^\circ$C in 20% (*m/w*) glycerol and were cultivated by the two-fold passage in Vegitone MRS broth modified (Sigma–Aldrich Co., Darmstadt, Germany), using 1% *v/v* of inoculum.

Bacterial cultures were centrifuged at $10,000\times\ g$ for 5 min. The cell pellets were obtained, washed in distilled water, and then resuspended in 10 mL of sterile 0.85% saline solution to their original volume. Then, prepared bacterial cultures were transferred to 300 g pasteurized pumpkin pulp with 17% *v/v* sucrose and were found to be at the level of 95; log, C.F.U. (colony forming units) ml$^{-1}$ in the case of each strain.

### 2.3. Functional Frozen Desserts Preparation

The pumpkin pulp preparation and the conditions of the fermentation process have been described in a previous study [16]. In this study, the potentially probiotic sorbets were produced in four formulations (Table 1) based on the author's Polish patent [17] and a previous study [9].

**Table 1.** Description of the investigated samples.

| Sample Code | Sample Name | Pumpkin Cultivar | Applied Bacterial Strain |
|---|---|---|---|
| MY | Pumpkin flesh | *"Melon Yellow"* | - |
| PPMY | Pumpkin pulp | *"Melon Yellow"* | - |
| MY1 | Fermented pumpkin pulp | *"Melon Yellow"* | *Lactobacillus rhamnosus* Lock 0900 |
| MY2 | Fermented pumpkin pulp | *"Miranda"* | *Lacticaseibacillus casei* O14 |
| SMY1 | Pumpkin sorbet | *"Melon Yellow"* | *Lactobacillus rhamnosus* Lock 0900 |
| SMY2 | Pumpkin sorbet | *"Melon Yellow"* | *Lacticaseibacillus casei* O14 |
| M | Pumpkin flesh | *"Miranda"* | - |
| PPM | Pumpkin pulp | *"Miranda"* | - |
| M1 | Fermented pumpkin pulp | *"Melon Yellow"* | *Lactobacillus rhamnosus* Lock 0900 |
| M2 | Fermented pumpkin pulp | *"Miranda"* | *Lacticaseibacillus casei* O14 |
| SM1 | Pumpkin sorbet | *"Miranda"* | *Lactobacillus rhamnosus* Lock 0900 |
| SM2 | Pumpkin sorbet | *"Miranda"* | *Lacticaseibacillus casei* O14 |

All of the fermented pumpkin mixtures, contained a concentration of 66.5% *w/v* of fermented pumpkin pulp, 17% *w/v* of sucrose and 1% *v/v* of inoculum, relative to the original value of final ice mass. The fermentation of the presented mixtures with selected strains was incubated at temperature 32 $^\circ$C for 26 h. Then, the semi-product, fermented pumpkin pulp was combined with other ingredients of the recipe and was cooled in laboratory terms.

Each batch of investigated products was produced in triplicates. The analysis of frozen desserts was performed 24 h after the manufacturing process.

### 2.4. Microbiological Analysis

The analyses were carried out by the TEMPO® System, an automated quality indicator solution (BioMérieux, Mercy Etoile, France). The calculation of LAB bacteria number in the investigated fermented pumpkin pulps and final products (log CFU $g^{-1}$) were according to the TEMPO® System. This system is based on the most probable number (MPN) method. The dilution of the samples was 1/400 in a single vial. The inoculated medium was moved into the Tempo card by Tempo Filler.

The Tempo LAB test was able to achieve performance levels that were comparable to the NF ISO15214:1998 standard [26], the cards were incubated at 37 °C for 48 h. The software system that decides which of the wells tested positive automatically analyzed the data. The number of positive wells obtained about the volume of wells and sample dilution allowed for automatic enumeration of the results in CFU $g^{-1}$.

### 2.5. Acidity Analysis (pH)

The pH values of functional frozen desserts were measured in triplicate using a pH meter (Elmetron, CP551, Zabrze, Poland). A product sample of 25 g was used for each measurement. The obtained results were read with an accuracy of 0.001.

### 2.6. Sugars Content

The sugar content in investigated samples was determined by the Lane–Eynon method, which can refer to Polish Standard PN-90/A-75101/07 [27].

The sucrose content was calculated according to the following Equation (1):

$$C_s = (Co - Cr) \times 0.95 \tag{1}$$

where: Co—means total sugar content, Cr—means reducing sugar content.

### 2.7. Total Carotenoids Content

The total carotenoid content in the raw material, fermented pumpkin pulps, and finally different formulations of sorbets was determined by spectrophotometric method according to a Polish Standard (PN-90 A-75101/12) [28]. It was measured in triplicates by using the spectrophotometer GenesysTM 20 (Thermo Scientific Co., Waltham, MA, USA) at wavelength = 450 nm. The obtained results are expressed as mg $\times$ 100 $g^{-1}$ FW.

### 2.8. Determination of Antioxidant Activity

#### 2.8.1. DPPH Radical Scavenging Assay

The antioxidant activity was measured according to DPPH (2,2-diphenyl-1-picrylhydrazyl) modified method Brand-Williams et al. (1995) [29]. Briefly, 25 g of investigated sample was mixed with 100 mL 0.1 mM DPPH (Sigma–Aldrich, Darmstadt, Germany) in ethanol. It was shaken, and then, after 24 h it was filtered. After incubation of the prepared solution for 30 min at room temperature in dark conditions, the absorbance was measured at 517 nm using a spectrophotometer GenesysTM 20 (Thermo Scientific Co., Waltham, MA, USA). The inhibition percentage of DPPH radical discoloration was calculated using the following Equation (2):

$$\% \text{ Inhibition of DPPH} = 100 \, (A0 - Ar)/A0 \tag{2}$$

where A0 is the absorbance of the control; Ar is the absorbance of the extract.

#### 2.8.2. ABTS Radical Scavenging Assay

The antioxidant capacity of investigated extracts by ABTS method was conducted according to Re et al. (1999) [30]. The ABTS radical cation (ABTS•+) was produced by the reaction of 7 mM ABTS (2,2′-azino-bis 3-ethylbenzothiazoline-6-sulphonic acid; Sigma–Aldrich, Darmstadt, Germany) solution with 2.45 mM potassium persulphate (Sigma–Aldrich, Darmstadt, Germany). The prepared mixture was kept in the dark for 12 h at ambient temperature before use. The ABTS solution was diluted with ethanol to obtain an

absorbance of 0.70 at 734 nm wavelength. Changes in the concentration of the ABTS•+ radical cations were determined by spectrophotometer GenesysTM 20 (Thermo Scientific Co., Waltham, MA, USA) after 6 min incubation with the investigated extracts. The ABTS radical scavenging activity was calculated using the following Equation (3):

$$\% \text{ Inhibition of ABTS} = 100 \, (A0 - Ar)/A0 \tag{3}$$

where A0 is the absorbance of the control; Ar is the absorbance of the extract.

### 2.9. Color Measurement

The Konica Minolta CM-2300d spectrophotometer (Konica Minolta Business Technologies, Inc., Osaka, Japan) was used to measure the CIE color parameters (L*, a* and b*) drawing upon the method described by Feistauer-Gomes et al. 2018 [31]. While Mendoza et al. 2006 [32] recommended this system as the best color space for measurement in foods with curved surfaces.

The spectrophotometer was calibrated with a white standard tile.

The relative color difference index (ΔE) was calculated according to the following Equation (4):

$$\Delta E = [(L_0 - L)^2 + (a_0 - a)^2 + (b_0 - b)]^{0.5} \tag{4}$$

where: parameter L—lightness coefficient (L = 0 indicates black and L = 100 indicates white [dimensionless value]; parameter a—red color coefficient [dimensionless value]; parameter b—yellow color coefficient [dimensionless value]; $L_0$, $a_0$, $b_0$—color coefficients relate to for flesh pumpkin for each cultivar, respectively [dimensionless value].

### 2.10. Sensory Evaluation

Quantitative descriptive analysis (QDA) following the ISO procedure [33] was performed to evaluate the sensory characteristics of the investigated pumpkin frozen desserts. The sensory estimation of products was carried out after 24 h, and the manufacturing process involving 10 staff members of the Department of Food Gastronomy and Food Hygiene, WULS. The panelists were trained with regard to the basic sensory evaluation method [34]. The experts were aged between 28 and 55, with good knowledge of sensory evaluation methods, including profiling the estimation of frozen desserts such as sorbets.

The assessed sensory characteristics of the products were previously selected and defined by a panel of trained panelists (experts). The following sensorial properties were ultimately assessed: density, smoothness, bitter taste, pungent taste, burning taste, pumpkin flavor, sweet flavor, acidic flavor, other flavor, tone of color, as well as overall quality.

The panelists determined the intensity of each of the quality attributes and conducted their assessment on an appropriate scale (linear graphical scale 0 (low)–10 (high) conventional units [c.u.].

Moreover, based on the above distinguishing attributes, the overall sensory quality of the frozen desserts was additionally determined using also such mentioned above, separate scale. The samples of pumpkin sorbets for the sensorial analysis were at 12 ± 1 °C.

### 2.11. Statistical Analysis

All measurements were performed in triplicates. The one-way analysis of variance (ANOVA) test was followed by the Fisher NIR test, with the overall significance level set at 0.05 using STATISTICA 13.3 PL software (StatSoft, Kraków, Poland).

The principal component analysis (PCA) method is used, among other things, to reduce the number of variables describing the phenomena or to discover the validity between the variables [35]. The PCA method was used to interpret the sensory, color evaluation, and the total carotenoid content results of investigated pumpkin frozen desserts.

## 3. Results

### 3.1. Microbiological Analysis and pH Changes

In the first part of the study, the pasteurized pumpkin pulps of two cultivars (Section 2; Table 1) were fermented with two potential probiotic bacterial strains *L. rhamnosus* Lock 0900 and *L. casei* O14.

The changes in the count of LAB bacteria and pH values during "*Melon Yellow*" and "*Miranda*" cultivars pumpkin pulps fermentation are shown in Tables 2 and 3.

**Table 2.** Changes in the count of LAB bacteria and pH values during pumpkin pulps fermentation (cultivar **"***Melon Yellow***"** of the *Cucurbita maxima* species).

| Sample Code | pH | Count of Bacteria [log CFU/g] |
|:---:|:---:|:---:|
| MY | 6.95 ± 0.03 [c] | - |
| PPMY | 6.92 ± 0.10 [c] | - |
| MY1 | 4.50 ± 0.03 [a] | 9.48 ± 0.07 [c] |
| MY2 | 4.53 ± 0.03 [a] | 9.41 ± 0.09 [c] |
| SMY1 | 4.57 ± 0.05 [a] | 8.69 ± 0.11 [a] |
| SMY2 | 4.71 ± 0.04 [b] | 8.82 ± 0.05 [b] |

Explanatory notes: The presented samples are coded according to Section 2; Table 1. The results are expressed as the mean ± standard deviation (*n* = 3). Values denoted by different letters differ significantly (*p* < 0.05).

**Table 3.** Changes in the count of LAB bacteria and pH values during pumpkin pulps fermentation (cultivar "*Miranda*" of the *Cucurbita moschata* species).

| Sample Code | pH | Count of Bacteria [log CFU/g] |
|:---:|:---:|:---:|
| M | 6.51 ± 0.03 [d] | - |
| PPM | 6.53 ± 0.07 [d] | - |
| M1 | 4.70 ± 0.09 [b] | 9.26 ± 0.12 [d] |
| M2 | 4.45 ± 0.08 [a] | 9.16 ± 0.09 [c] |
| SM1 | 4.80 ± 0.03 [c] | 8.35 ± 0.10 [a] |
| SM2 | 4.61 ± 0.03 [b] | 8.94 ± 0.04 [b] |

Explanatory notes: The presented samples are coded according to Section 2; Table 1. The results are expressed as the mean ± standard deviation (*n* = 3). Values denoted by different letters differ significantly (*p* < 0.05).

During the fermentation, as a result of the metabolic activity of the used bacterial strains, a decrease in pH values in each sample was observed. The pH of the fermented mass dropped from the initial 6.92 in the case of the cultivar "*Melon Yellow*" of the *Cucurbita maxima* species to 4.5–4.53, respectively, and in the case of the cultivar "*Miranda*" of the *Cucurbita moschata* species from 6.53 to the 4.45 to 4.70, respectively (Tables 2 and 3).

The significantly highest count of LAB bacteria (*p* < 0.05) among all the fermented samples of pumpkin pulps was observed in the cultivar "*Melon Yellow*" of the *Cucurbita maxima* species at 9.16–9.25 CFU/g, respectively (Table 2).

It has been observed that regardless of the pumpkin cultivar used, pulps fermented with the strain *Lacticaseibacillus casei* O14 are characterized by a significantly higher count of LAB bacteria compared to the samples fermented with the other strain, *L. rhamnosus* Lock 0900 (Tables 2 and 3).

As illustrated in Figure 1, a significantly higher count of LAB bacteria was also obtained in samples of sorbet produced with the cultivar "*Melon Yellow*" of the *Cucurbita maxima* species, SMY1 (8.81 CFU/g) (fermented with *Lactobacillus rhamnosus* Lock 0900) and SMY2 (8.95 CFU/g) (fermented with *Lacticaseibacillus casei* O14), respectively. At the same time, this corresponds to significantly lower pH values (Figure 1) compared to samples SM1 and SM2.

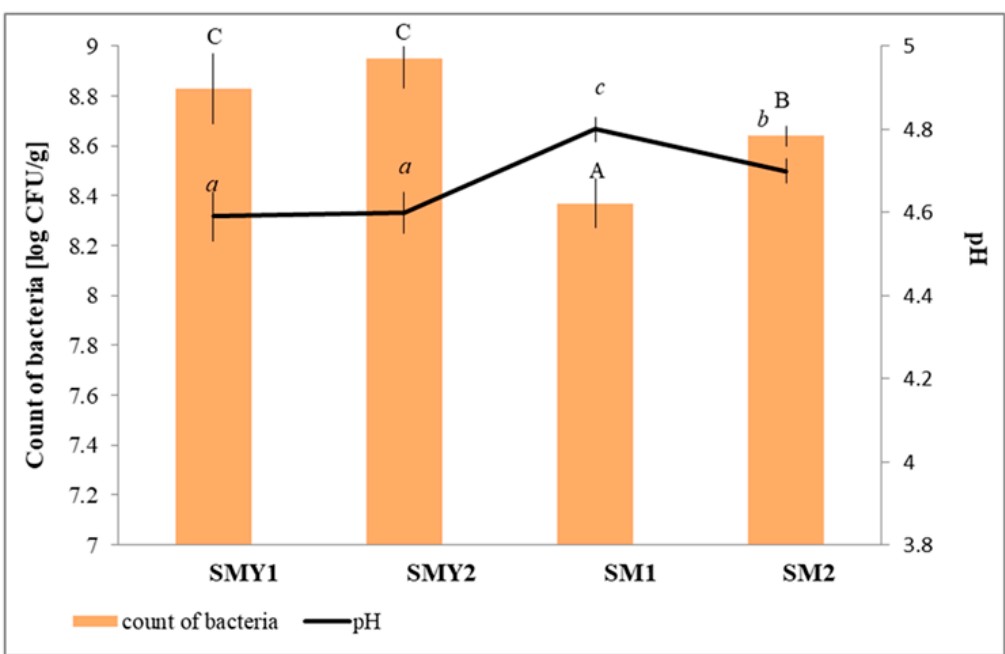

**Figure 1.** Impact of pumpkin cultivar used and bacterial strain on pH value and the count of LAB bacteria in investigated functional frozen desserts. Explanatory notes: The presented samples are coded according to Section 2; Table 1. The results are expressed as the mean ± standard deviation (*n* = 3). Mean values denoted by the same letters (upper case letters—in regard LAB count, and lower-case letters—in regard to pH value) do not differ significantly (*p* > 0.05).

The bacterial strains used in the study significantly varied the sorbet group based on the cultivar "*Miranda*" of the *Cucurbita moschata* species, both in terms of pH value and count of LAB bacteria. In the sample SM2 (Figure 1) the highest count of LAB bacteria 8.95 CFU/g and the lowest pH value of 4.7 were noted.

### 3.2. Sugars Content

The changes in the total sugar content, reducing sugars during fermentation, and frozen desserts manufactured with different pumpkin cultivars are shown in Table 4 ("*Melon Yellow*" cultivar) and Table 5 ("*Miranda*" cultivar).

**Table 4.** Changes in the total sugars content, reducing sugars during fermentation and frozen desserts manufacture (cultivar "*Melon Yellow*" of the *Cucurbita maxima* species).

| Sample Code | Reducing Sugars [%] | Total Sugars [%] | Saccharose [%] |
|:---:|:---:|:---:|:---:|
| MY | 2.30 ± 0.07 [e] | 3.12 ± 0.10 [d] | 0.78 ± 0.06 [c] |
| PPMY | 2.11 ± 0.10 [d] | 2.56 ± 0.05 [c] | 0.44 ± 0.06 [b] |
| MY1 | 1.43 ± 0.06 [b] | 1.84 ± 0.05 [b] | 0.35 ± 0.06 [a] |
| MY2 | 1.28 ± 0.05 [a] | 1.75 ± 0.11 [a] | 0.45 ± 0.07 [b] |
| SMY1 | 1.39 ± 0.09 [b] | 15.20 ± 0.08 [e] | 13.10 ± 0.10 [d] |
| SMY2 | 1.31 ± 0.09 [a] | 15.62 ± 0.05 [f] | 13.50 ± 0.07 [e] |

Explanatory notes: The presented samples are coded according to Section 2; Table 1. The results are expressed as the mean ± standard deviation (*n* = 3). Values denoted by different letters differ significantly (*p* < 0.05).

The significantly highest total sugar content was noted (3.12%) in the pumpkin flesh cultivar "*Melon Yellow*" of the *Cucurbita maxima* species. As a result of the fermentation of the pumpkin pulps of both plant cultivars, the reducing sugars, total sugars, and sucrose content (Table 4) have been reduced. In the case of the cultivar "*Melon Yellow*" of the *Cucurbita maxima* species, during the fermentation process, the reducing sugars content

was decreased by 38% (PMY1) and 44% (PMY2), respectively, compared to raw matter. This was related to the high metabolic activity and number of LAB bacteria noted in these semi-products (Figure 1). However, in pumpkin pulps with the cultivar "*Miranda*" of the *Cucurbita maxima* species, the reducing sugars content decreased by 36% (PM1) and 40% (PM2), respectively, relative to raw pumpkin flesh. A combination of all components of the recipe, including the addition of sucrose, showed an increase in total sugars and saccharose content in the final pumpkin sorbets (Table 5).

**Table 5.** Changes in the total sugars content, reducing sugars during fermentation and frozen desserts manufacture (cultivar "*Miranda*" of the *Cucurbita moschata* species).

| Sample Code | Reducing Sugars [%] | Total Sugars [%] | Saccharose [%] |
|---|---|---|---|
| M | 1.25 ± 0.07 [d] | 2.05 ± 0.06 [d] | 0.76 ± 0.04 [c] |
| PPM | 1.00 ± 0.03 [c] | 1.80 ± 0.03 [c] | 0.76 ± 0.11 [c] |
| M1 | 0.80 ± 0.12 [b] | 1.45 ± 0.06 [b] | 0.62 ± 0.05 [b] |
| M2 | 0.75 ± 0.06 [a,b] | 1.31 ± 0.05 [a] | 0.53 ± 0.07 [a] |
| SM1 | 0.76 ± 0.10 [a,b] | 12.10 ± 0.06 [f] | 10.77 ± 0.12 [e] |
| SM2 | 0.69 ± 0.09 [a] | 11.20 ± 0.10 [e] | 9.98 ± 0.07 [d] |

Explanatory notes: The presented samples are coded according to Section 2; Table 1. The results are expressed as the mean ± standard deviation (*n* = 3). Values denoted by different letters differ significantly (*p* < 0.05).

However, the used pumpkin cultivar still differed from the investigated samples in terms of total sugar content. Products (SMY1, SMY2) were characterized by significantly higher total sugar content 15.20–15.62% among all sorbet samples. Whilst it also correlated to the general higher total sugar content in the pumpkin flesh of the "*Melon Yellow*" cultivar of the *Cucurbita maxima* species.

## 3.3. Determination of Antioxidant Activity

The antioxidant characteristics of investigated raw materials, fermented pumpkin pulps and functional frozen desserts are presented in Figure 2a,b.

The DPPH activity of fresh pumpkin pulp referred to as % DPPH inhibition ranged from 9.50 % in the variety "*Miranda*" of the *Cucurbita moschata* species to 18.0% in the variety "*Melon Yellow*" of the *Cucurbita maxima* species, respectively (Figure 2a,b).

As a result of the lactic fermentation process, regardless of the used bacterial strain, the antioxidant activity of the fermented pulps was approximately three times higher for the cultivar "*Melon Yellow*" of the *Cucurbita maxima* species and two times higher for the cultivar "*Miranda*" of the *Cucurbita moschata* species compared to the raw material.

In the final products, these values have not changed significantly. The bacterial strain used for fermentation did not significantly vary the pulps and frozen desserts of the same cultivar in terms of antioxidant capacity detected by DPPH and ABTS assay.

The obtained results of antioxidant capacity detected by ABTS assay were slightly higher for the investigated raw material of two cultivars and all pumpkin pulps and sorbets compared to that by DPPH assay (Figure 2a,b).

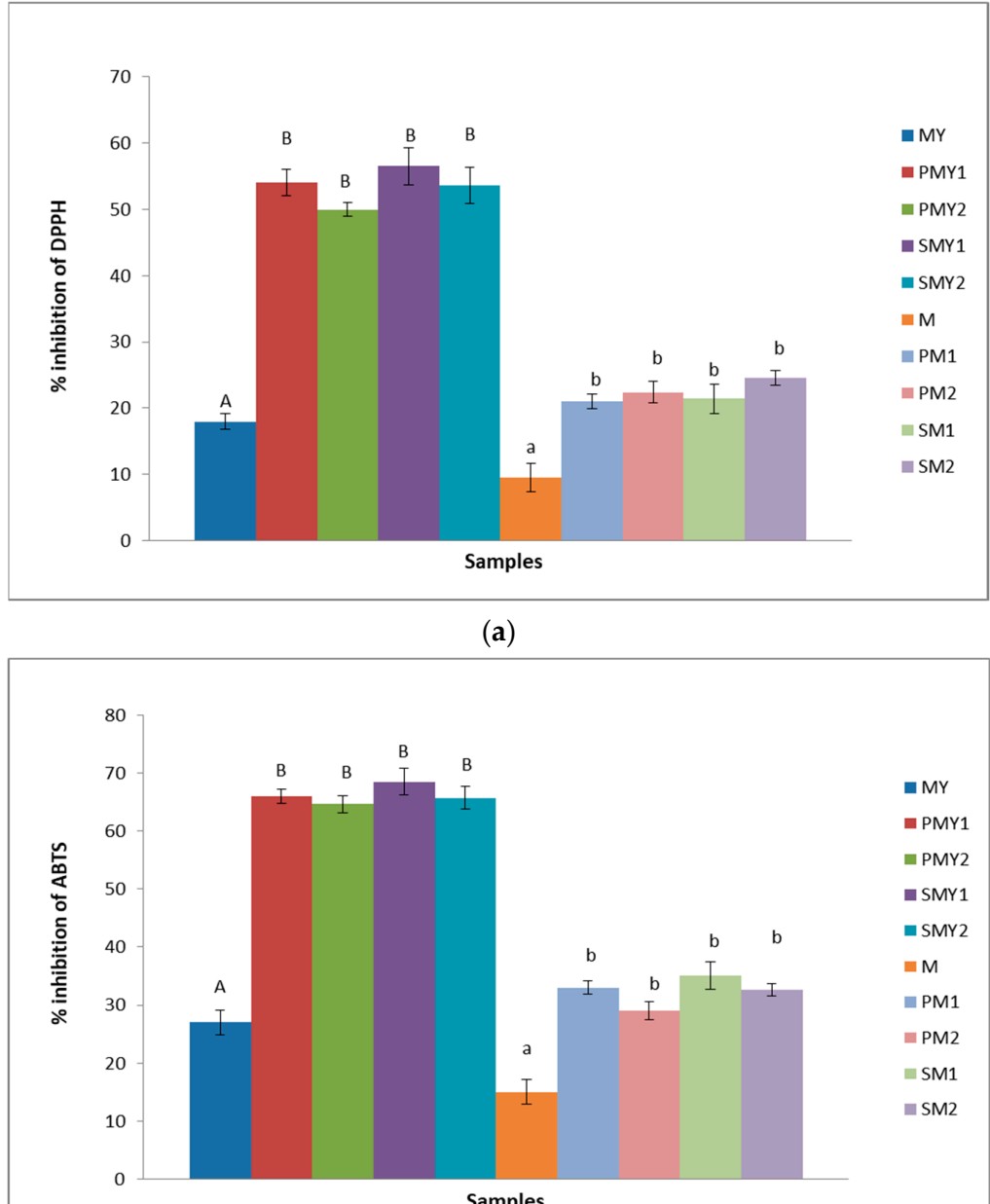

(**a**)

(**b**)

**Figure 2.** (**a**) Antioxidant characteristics of investigated samples. DPPH Method; (**b**) Antioxidant characteristics of investigated samples. ABTS Method. Explanatory notes: The presented samples are coded according to Section 2; Table 1. Values denoted by different upper-case letters in the cultivar "*Melon Yellow*" of the *Cucurbita maxima* species raw material, pumpkin fermented pulp and frozen desserts, differ significantly ($p < 0.05$). Values denoted by different lower-case letters in the "*Miranda*" of the *Cucurbita moschata* species raw material, pumpkin fermented pulp and frozen desserts, differ significantly ($p < 0.05$). The results are expressed as the mean $\pm$ standard deviation ($n = 3$).

### 3.4. Color Measurement and Total Carotene Content

The changes in total carotenoids content and color values (L* a* b*) in the investigated pumpkin flesh, pumpkin pulps and probiotic frozen desserts are shown in Tables 6 and 7.

**Table 6.** Changes in total carotenoids content (on fresh weight of tissue -FW) and color values (L* a* b*) in investigated pumpkin pulps and probiotic frozen desserts (cultivar "*Melon Yellow*" of the *Cucurbita maxima* species).

| Sample Code | Total Carotenoids (μg/g FW) | L* | a* | b* | ΔE |
|---|---|---|---|---|---|
| MY | 4.30 ± 0.65 [e] | 47.01 ± 2.10 [b] | 10.96 ± 1.14 [c,d] | 45.53 ± 0.15 [d] | - |
| PPMY | 3.90 ± 0.41 [e] | 45.91 ± 1.03 [b] | 12.49 ± 1.12 [d] | 47.82 ± 2.10 [d,e] | 2.96 |
| MY1 | 2.90 ± 0.11 [c] | 41.77 ± 1.23 [a] | 12.58 ± 0.09 [d] | 43.95 ± 1.11 [d] | 5.74 |
| MY2 | 3.20 ± 0.09 [d] | 42.60 ± 1.51 [a] | 10.24 ± 0.02 [c] | 26.75 ± 0.04 [b] | 19.30 |
| SMY1 | 1.80 ± 0.09 [a] | 50.34 ± 0.34 [c] | 9.92 ± 0.12 [a, b] | 35.68 ± 0.81 [c] | 10.45 |
| SMY2 | 2.60 ± 0.03 [b] | 54.25 ± 0.23 [d] | 4.93 ± 0.15 [a] | 20.16 ± 0.13 [a] | 27.06 |

Explanatory notes: The presented samples are coded according to Section 2; Table 1. The results are expressed as the mean ± standard deviation (*n* = 3). Values denoted by different letters differ significantly (*p* < 0.05).

**Table 7.** Changes in total carotenoids content (on fresh weight of tissue -FW) and color values (L* a* b*) in investigated pumpkin pulps and probiotic frozen desserts (cultivar "*Miranda*" of the *Cucurbita moschata* species).

| Sample Code | Total Carotenoids (μg/g FW) | L* | a* | b* | ΔE |
|---|---|---|---|---|---|
| M | 1.75 ± 1.42 [e,f] | 44.5 ± 2.04 [c] | 14.01 ± 1.23 [e] | 39.65 ± 0.91 [d] | - |
| PPM | 1.50 ± 0.35 [e] | 43.81 ± 1.78 [c] | 12.21 ± 0.01 [d] | 41.50 ± 1.11 [d] | 2.67 |
| M1 | 1.00 ± 0.10 [c] | 40.00 ± 0.11 [a] | 8.13 ± 0.12 [c] | 34.37 ± 0.09 [c] | 9.15 |
| M2 | 1.15 ± 0.08 [d] | 42.54 ± 0.54 [b] | 7.50 ± 0.07 [b] | 33.24 ± 0.06 [b] | 9.34 |
| SM1 | 0.65 ± 0.12 [a] | 50.75 ± 1.39 [d] | 3.26 ± 0.09 [a] | 32.45 ± 0.09 [a] | 14.40 |
| SM2 | 0.95 ± 0.11 [b] | 50.10 ± 2.33 [d] | 3.21 ± 0.11 [a] | 34.20 ± 0.11 [c] | 13.33 |

Explanatory notes: The presented samples are coded according to Section 2; Table 1. The results are expressed as the mean ± standard deviation (*n* = 3). Values denoted by different letters differ significantly (*p* < 0.05).

The pumpkin flesh presented the highest total carotenoids contents. It ranged from 1.75 μg/g FW (the cultivar "*Miranda*" of the *Cucurbita moschata* species) to 4.30 μg/g FW (the cultivar "*Melon Yellow*" of the *Cucurbita maxima* species) (Tables 6 and 7).

As a result of the heat treatment during the production of the pumpkin pulps, the total carotenoids content has decreased slightly compared to the raw material content, irrespective of the variety of the used plant (*p* < 0.05). Further technological processes, such as the fermentation of purees and the production of sorbet, also resulted in a significant reduction in total carotenoids content for both pumpkin cultivars (*p* < 0.05) (Tables 6 and 7).

Depending on the pumpkin cultivar, total carotenoid content was recorded in frozen sorbets at a level of 0.65 to 0.95 μg/g FW (cultivar "*Miranda*" of the *Cucurbita moschata* species) and 1.80 to 2.60 μg/g FW, respectively.

The color characteristics of pumpkin flesh cultivars were closer to each other, as yellowness (b*) was dominant in comparison to redness (a*) (Tables 6 and 7).

During fermentation of pumpkin pulps with selected bacterial strains (Tables 6 and 7), a significant reduction in value L* was observed, which indicates the lightness of samples relative to the raw material, irrespective of the used plant cultivar. The samples were slightly dark. However, in the final products, a significant increase in L* values regarding fermented pumpkin pulps was observed.

The values of b* for the investigated sorbets are in the positive range, indicating the presence of yellow in the products, originating from the base components of the frozen desserts. The b* values decreased during the lactic acid fermentation of pumpkin pulps and the manufacturing of potentially functional sorbets.

In the case of the cultivar "*Miranda*" of the *Cucurbita moschata* species, the value of parameter a\* represents the red color coefficient (Table 7), and was significantly reduced by lactic acid fermentation and sorbet production. In the samples based on the cultivar "*Melon Yellow*" of the *Cucurbita maxima* species, however, the value of parameter a\* decreased significantly in only one sample of the fermented pulps (MY2) and in the final products (SMY1, SMY2) (Table 6).

The pumpkin color changes as a result of pumpkin pulp production, lactic acid fermentation, and sorbet manufacture process describe by an increase in the total color difference parameter (ΔE). The maximum ΔE value was calculated for sample SMY2 based on the pumpkin pulp cultivar "*Melon Yellow*" of the *Cucurbita maxima* species, fermented with *L. casei* O14. On the other hand, the minimum ΔE value was noted in the case of SMY1 sample (Table 6) manufactured with the same pumpkin cultivar, but with another bacterial strain—*L. rhamnosus* Lock 0900.

Lastly, it should be noted that ΔE values of products based on the cultivar "*Miranda*" of the *Cucurbita moschata* species (SM1, SM2), were close to each other (Table 7), in contrast to products based on the cultivar "*Melon Yellow*" of the *Cucurbita maxima* species (SMY1, SMY2) (Table 6).

### 3.5. Sensory Characteristics of Functional Frozen Dessert

The results of sensory evaluation of the investigated samples of pumpkin probiotic frozen desserts are shown in Figure 3. The obtained sensory profiles of these treatments were different depending on the pumpkin cultivar used.

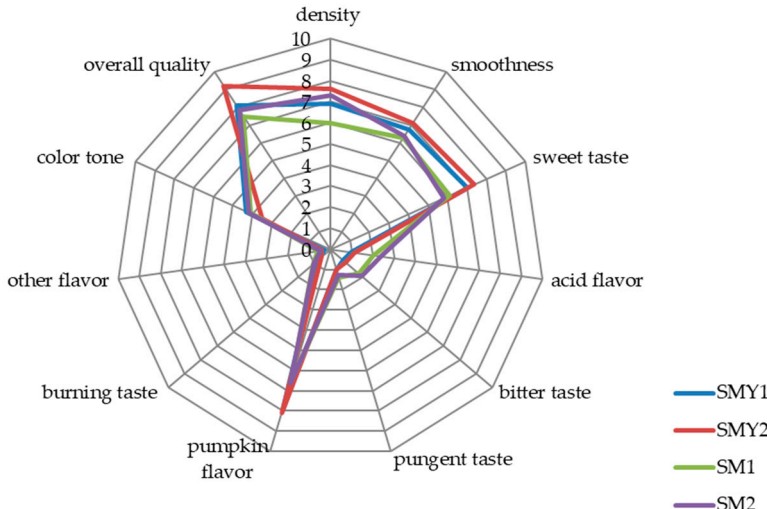

**Figure 3.** Sensory profile of functional frozen desserts manufactured by using different pumpkin cultivars. Explanatory notes: The presented samples are coded according to Section 2; Table 1.

The treatments SMY1, SMY2 (Section 2, Table 1) manufactured with the cultivar "*Melon Yellow*" of the *Cucurbita maxima* species were characterized by a high intensity of sweet taste (7.05–7.38 c.u.), pumpkin flavor (7.05–7.38 c.u.) and the highest notes of texture attribute-smoothness (6.75–7.1 c.u.). All treatments of the investigated products had a good overall quality (mean higher than 7.5 c.u.). The sample SMY2 (Section 2, Table 1) revealed the highest intensity of overall quality (9.2 c.u.), which can be attributed to the high intensity of positive sensory factors mentioned above (Figure 3). However, the negative sensory factors such as acid flavor, pungent taste and bitter taste were noted to be more intense in the case of pumpkin sorbets based on the other plant cultivar "*Miranda*" of the *Cucurbita moschata* species.

The PCA score plot (Figure 4) also showed clear differences according to the pumpkin cultivar used in the manufacture of potentially probiotic frozen desserts. Therefore, the selected parameters of sensory quality, the relative color difference index (ΔE)

and total carotenoids content in the investigated final products are greatly affected by pumpkin cultivar.

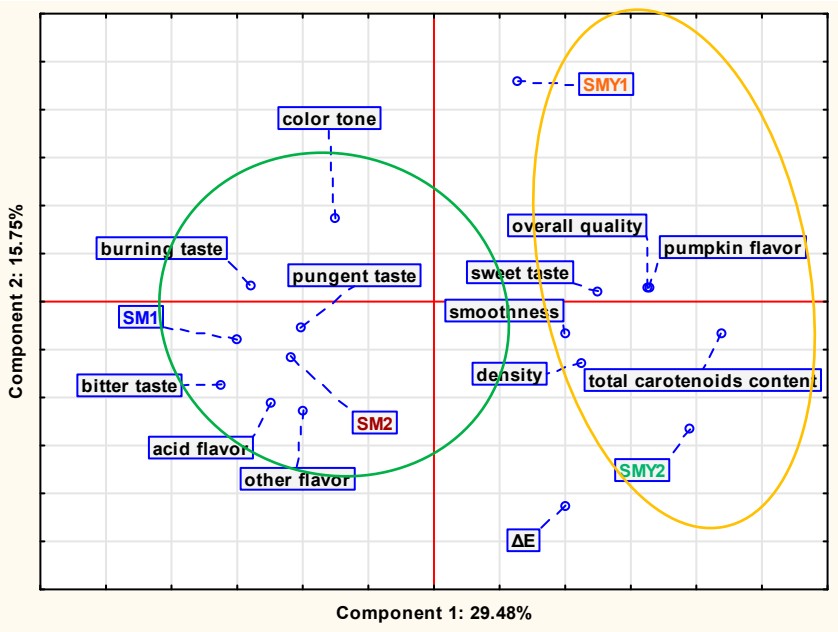

**Figure 4.** Principal component analysis (PCA) graph of the selected sensory attributes, total carotenoids content and parameters of color measurement in the investigated samples of functional pumpkin frozen desserts. Explanatory notes: The presented samples are coded according to Section 2; Table 1. The abbreviations: ΔE, L*, a*, and b* correspond to color components.

In this study, three principal components were identified, of which the total of the components explains 55.9% of the variance of variables. The sum of the first two main principal components was used for the analysis of the obtained results in this study. A multivariate analysis of the data (Figure 4) showed that 45.23% of the variation between the samples was explained in the first two principal components.

The first component explained 29.48% of the total variability and was related to smoothness, sweet taste, pumpkin flavor, density, total carotenoids content, ΔE, other flavor and overall quality. However, the second component explained 15.75% of the total variability and was related to bitter taste, pungent taste, acid flavor, burning taste and color tone.

Samples of functional frozen pumpkin desserts were grouped into two clusters with their sensory attributes, changes in color, carotenoids content and selected pumpkin cultivar. The first group constituted two samples (SMY1, SMY2) of products based on the cultivar "*Melon Yellow*" of the *Cucurbita maxima* species. These samples group is located the right side of the plot. The attributes located in this area of the plot indicate that the investigated frozen desserts (SMY1, SMY2) have quality similarities, such as the highest intensity of overall acceptance, sweet taste, pumpkin flavor and the highest notes of smoothness (Figure 4). The second group constituted SM1, SM2 products based on the cultivar "*Miranda*" of the *Cucurbita moschata* species, and they are located in the lower and left parts of the presented plot (Figure 4). These samples were characterized by high intensity of bitter taste, acid flavor and another flavor.

## 4. Discussion

The functional food concept is still the current trend in developing novel food technology, as nutrition plays an important role in improving well-being. The Functional Food Center (FFC) defines functional food as "natural or processed foods that contain known or unknown biologically-active compounds; the foods, in defined, effective, and non-toxic

amounts, provide a clinically proven and documented health benefit for the prevention, management, or treatment of chronic disease" [36]. These biologically-active compounds can be obtained from various sources, e.g., vitamins, minerals, antioxidant compounds, polyphenols, prebiotics, or probiotics.

Non-dairy frozen desserts are feasible food matrices for the incorporation of probiotics. However, in the case of plant-based frozen desserts, raw material, ingredients of the recipe and stages of the manufacture process can significantly affect the viability of probiotic bacteria [37].

According to Dronkers et al. (2020) [38], viability is a crucial requirement to meet the definition of a probiotic. The probiotic efficacy is dependent upon delivering an adequate dose throughout the product's shelf life. Probiotic cells must survive passage through the gastrointestinal tract, reach the colon in sufficient numbers, and finally adhere to and colonize the gut epithelium. Probiotic dosages typically range between $10^7$ and $10^{11}$ colony-forming units (CFU)/day [39,40].

Genevois et al. (2016) [41] suggested that *Lactobacillus* strains may metabolize and adhere to pumpkin's polysaccharides. In this study, pumpkin pulps fermented with probiotic bacteria as semi-product for functional frozen desserts manufacture were used. In all investigated sorbets the numbers of LAB bacteria were detected as higher than 8.30 log CFU /g (Figure 1). These findings are in agreement with other authors [42–44] and suggest that one portion of pumpkin sorbet can serve as a therapeutic dose of probiotics.

For example, Krawęcka et al. (2021) [42] reported that the count of probiotic LAB strain *Lactiplantibacillus plantarum* 299v in fresh non-fermented avocado ice cream ranged from 3.9 CFU/g to 8.1 CFU/g, respectively, depending on the method of inoculum application. On the other hand, Väkeväinen et al. (2021) [43] observed counts of 6.5 CFU/g and 7.5 log CFU/g for the *L. plantarum* Q823 in a frozen functional vegan blackcurrant product with quinoa fermented base. In turn, Da Silva Machado et al. (2021) [44] reported the stability of the jucara and banana sorbets, with probiotic viability above 7.9 log CFU/mL.

In the present study, we observed higher counts of LAB bacteria in sorbet samples (SMY1; SMY2) based on fermented pumpkin pulps of the cultivar "*Melon Yellow*" of the *Cucurbita maxima* species. The fact can be explained by the higher total sugar content in this plant cultivar (Table 4). It is important to observe that after the fermentation process, LAB count increased, while pH value and sugar content decreased. However, within a particular variety, we observed a tendency related to a significantly higher count of LAB bacteria in products fermented with bacterial strain *L. casei* O14, which was isolated from the plant-origin food matrix [23] in comparison to human origin *L. rhamnosus* Lock 0900. This observation can be important for product development strategies. The selection of appropriate probiotic strains should take into account both functional properties for human health and technological suitability, such as good survivability and resistance to harsh technological conditions. Moreover, searching for the best strain of probiotic bacteria should probably include the source of isolation.

The fortification of "functional food matrices" with probiotics is the case of changes in bioactive components of products due to metabolic probiotic activity [45]. Antioxidants (carotenoids and phenolic compounds) quench the free radicals and protect against structural damage and dysfunction of organism cells, which can cause harmful health effects such as cancer and cardiovascular diseases. According to Saini et al. (2015) [46], the most significant aspect of carotenoids in the diet is the antioxidant and provitamin A activity and giving the food color. Processing and other post-harvest technologies have a considerable impact on the composition and bioavailability of carotenoids in foods.

In the present study, the pumpkin flesh and non-fermented pumpkin pulp presented the highest total carotenoid contents. It ranged from 1.50 μg/g FW (cultivar "*Miranda*" of the *Cucurbita moschata* species) to 4.30 μg/g FW (cultivar "*Melon Yellow*" of the *Cucurbita maxima* species) (Tables 6 and 7). The heat treatment of raw material and the production of pumpkin pulps did not cause any significant changes in the total carotenoid content. By contrast, Nawirska-Olszańska et al. (2011) [47] found that in pumpkin puree, the

other cultivar "*Karovita*" of the *Cucurbita maxima* species, a total carotenoid content of 7.4 µg /g FW. Quantitative and qualitative differences in the composition of carotenoids in fruit of the *Cucurbita maxima* and *Cucurbita moschata* were also shown by Azevedo-Meleio and Rodriguez-Amaya (2007) [48], indicating the genetic factor of this differentiation.

Provesi et al. (2011) [49] noted that as a result of the production process of pumpkin puree cultivar "*Exposição*" of the *Cucurbita maxima* species, there was a significant decrease in carotenoids content (such as: luteine, α-carotene). However, the content of cis-β-carotene increased fivefold.

In the present study, the process of lactic acid fermentation of pumpkin pulps with bacterial strains, *L. rhamnosus* Lock 0900 and *L. casei* O14, and the manufacture of the final products caused a significant decrease in total carotenoids content (Tables 6 and 7) compared to the raw material. In frozen sorbets, depending on the cultivar of the pumpkin total carotenoids content at the level 0.65–0.95 µg/g FW (cultivar "*Miranda*" of the *Cucurbita moschata* species) and 1.80–2.60 µg/g FW (cultivar "*Melon Yellow*" of the *Cucurbita maxima* species) was noted. It was noted that lower carotenoid content was related to the color values L* and a*. The color values L* (lightness) of raw material, fermented pumpkin pulps and final products were correlated ($p < 0.05$) negatively with total carotenoids (r = −0.48 for batches based on cultivar "*Melon Yellow*" of the *Cucurbita maxima* species; r = −0.85 for batches based on the cultivar "*Miranda*" of species the *Cucurbita moschata* species) in all pumpkin sorbets (data not shown). According to Itle and Kabelka (2009) [50], a negative correlation between L* and certain carotenoids would be expected because any increase in pigment would increase the darkness and thereby decrease L*.

As reported by Mapelli-Brahm et al. (2020) [51], the effect of lactic acid fermentation on the carotenoid content depends on some factors such as the matrix, carotenoid composition, and LAB strain. The fermentation does not cause important losses of carotenoids. However, the impact of fermentation processes on the carotenoid content in foods and their bioavailability is not yet well understood.

As shown in Tables 6 and 7, the used pumpkin cultivar significantly differentiated raw flesh in terms of total carotenoid content. The cultivar "*Melon Yellow*" of the *Cucurbita maxima* species was characterized by more than triple the content of those compounds. It was comparable with the results obtained from previous research by Nawirska-Olszańska, 2011 [52].

As a result of the fermentation process, regardless of the cultivar and bacterial strain used, the total carotenoid content slightly decreased. However, a higher loss rate was noted in samples fermented with the usage of *L. rhamnosus* Lock 0900. Regardless of the pumpkin cultivar used, the fermentation process resulted in a significant decrease in carotenoid content in the pumpkin pulps. Among all final products, the highest content of total carotenoids was reported for the sample SMY2 (Table 6).

On the other hand, Hassan and Barrakat (2018) [21] found that, depending on the level of addition of pumpkin pulps based on pumpkin fruits of the American pumpkin (*Cucurbita moschata* L.), the total carotenoid content ranged from 13.5 to 23.2 µg/g in ice cream.

The total carotenoid content may affect the antioxidant activity of pumpkin pulp products. However, considering the many variables which can influence the antioxidant activity results in food products, it is important to apply assays with different reaction mechanisms [53]. Therefore, in our study, due to the applied complex matrix such as vegetable sorbets, the antioxidant activity was determined by both DPPH and ABTS methods.

The previous studies [54,55] demonstrated that pumpkin cultivars can vary considerably in terms of antioxidant activity, which is the result of a large variation in the content of bioactive compounds. Results obtained in our study show that fermentation improved the ABTS and DPPH antioxidant capacity in probiotic frozen desserts (Figure 2).

Wu et al. (2020) [56] also reported that the antioxidant activities based on DPPH method were enhanced during fermentation of apple juice with addition of selected bacte-

rial strains of LAB. The authors suggested that the improvement of antioxidant capacity of fermented products was related to the increase in caffeic acid and phlorizin contents.

The antioxidant activities in the investigated functional frozen desserts cannot be attributed to their total carotenoids content (Tables 6 and 7), which was clearly decreased as a result of lactic acid fermentation. The improvement the ABTS and DPPH antioxidant capacity in fermented pumpkin pulps and frozen desserts can probably be associated with the fermentation process and actions of other antioxidant compounds such as phenolic compounds present in the fermented semi product—pumpkin puree and their mutual interactions. Some authors [57,58] indicated that phenolic and flavonoid compounds are responsible for the antioxidant activity of the pumpkin extracts. Investigating these issues will be certainly the aim of our further study.

The prepared pumpkin sorbets can be recognized as a functional food, taking into account the high addition of live probiotic bacteria, as well as high antioxidant activity and carotenoid compounds.

An important factor which cannot be ignored in food product development strategies is the assessment of sensorial attributes, which may be important from the consumer point of view. Lactic acid fermentation by modifying the chemical composition of the raw materials affects the sensory properties of food [59]. Taking into consideration the technological aspects, the use of a fruit addition in the manufacture of frozen desserts results in favorable changes to certain sensory qualities such as taste, texture or color [60]. In the case of vegetables, the yellowness and redness are attributed to the presence of carotenes [61]. In the presented study, it was found that color properties of the investigated pumpkin sorbets were significantly affected by the used semi-product, fermented pumpkin pulp (Tables 6 and 7).

The changes observed in ΔE in pumpkin pulps PPMY, PPM (Tables 6 and 7) were slight but could be due to boiled water replacing the intercellular air during processing steps and cellular content leaking into the water due to rupture of the cell membrane, as explained by Turkmen et al., 2006 [62].

The limits of total color difference (ΔE) in pumpkin pulp, after fermentation pulps and finally frozen desserts were also changed. As a result of the technological thermal process of pumpkin pulps production, slight changes in color compared to the flesh pumpkin were observed, at levels 2.67 (MY) and 2.96 (M) (Tables 6 and 7), respectively. They were unnoticeable even for the inexperienced observer (ΔE < 5). However, regarding the color difference (ΔE) parameter, the probiotic sorbets presented values from 13.33 to 27.06, respectively, which pointed to the large variations in total color, noticeable by consumers [63].

In this study, the "pumpkin flavor" and "yellow color" of non-dairy sorbets were affected by the intense yellow color of these unique fruits. Results of the sensory analysis are consistent with those reported by other authors, who observed that a higher level of sweetness in ice cream will provide higher acceptance and overall quality of these products [64]. Thus, implementation of the cultivar "*Melon Yellow*" of the *Cucurbita maxima* species into the recipe of frozen desserts with initial higher sugar levels guarantees a final product that will be rated highly in the aspect of overall quality.

According to Sipplee al. (2022) [65], base (dairy vs. plant), an "all-natural" and short ingredient list, and sweetener-related claims (naturally sweetened, reduced sugar, no added sugar) are the top attributes that contributed to the perception of a "healthier" frozen dessert.

It was found that our products contained a lower total sugar content compared to the pumpkin ice cream available on the market [66], probably due to the technology of sorbet manufacture using a semi-product fermented with probiotics plant base.

## 5. Conclusions

The most important conclusion of this study is that both the pumpkin cultivar and bacterial strain used for lactic acid fermentation significantly influence the quality of functional

plant-based frozen desserts. The differences in quality of the investigated products are noticeable in microbiological and sensory quality, color, and amount of bioactive compounds. Based on the results obtained in this study, it can be concluded that the cultivar "*Melon Yellow*" of the *Cucurbita maxima* species represents better suitability for functional sorbet production than the "*Miranda*" cultivar, taking into account probiotic survivability, antioxidant activity, as well as the overall sensory quality of a final product. Moreover, it was found that the potential probiotic strain of bacteria (*Lacticaseibacillus casei* O14) of plant-origin was characterized by better fermentation properties of pumpkin pulp than a human-origin probiotic strain of *Lactobacillus rhamnosus* Lock 0900. Therefore, the proper selection of bacterial strain can be a crucial step in fermented food product development.

The results obtained in this study may contribute to optimizing the formulation of functional foods, for example, non-dairy frozen desserts, taking into account the choice of the appropriate cultivar of plant-based material used, as well as a strain of bacteria for the fermentation process, which can guarantee a high overall quality of a final product.

**Author Contributions:** A.S.: Conceptualization, Supervision, Methodology, Software, Validation, Formal analysis, Investigation, Writing—review and editing, Visualization, Writing—original draft, Data Curation. D.Z.: Formal analysis, Investigation, Visualization, Data Curation. D.K.-K.: Formal analysis, Writing—review and editing. All authors have read and agreed to the published version of the manuscript.

**Funding:** This research received no external funding.

**Institutional Review Board Statement:** Not applicable.

**Informed Consent Statement:** Not applicable.

**Data Availability Statement:** Not applicable.

**Acknowledgments:** The Authors would like to thank the following Chairman of the Board of Organic Farming Cooperative "Dolina Mogilnicy" (Wołkowo, Poland) Roman Bartkowiak for helping with supply of plant material for research purposes.

**Conflicts of Interest:** The authors declare no conflict of interest.

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
