# Peer review of "Effect of Pumpkin Cultivar on the Selected Quality Parameters of Functional Non-Dairy Frozen Desserts"

_applsci, doi:10.3390/app12168063_

Round 1

Reviewer 1 Report

In this study, the authors investigated functional frozen desserts using different pumpkin cultivar. It would be great interested in the readers of the Applied Sciences. I only have a few questions: 

1. What is the definition of burning taste? How could it be coming from?

2. How does this type of product comparable to a commercial product? 

Author Response

We would like to thank the Reviewer for careful and thorough reading of this manuscript and for the thoughtful comments and constructive suggestions, which help to improve the quality of this manuscript. 

Reviewer 2 Report

What do you mean by used pumpkin cultivars in this line?

“This study was conducted to investigate the influence of used pumpkin cultivar as a 10 fermented semi-product on the selected quality parameters of functional non-dairy frozen desserts, 11 which were prepared using potentially probiotic strains cultures: L. rhamnosus Lock 0900 and L. 12 casei O14”.

Author should add some parameters from methodology along with statistical analysis in abstract section.

Add mean values or best results in the abstract quantitatively.

In second line of introduction you have mention “These factors contribute 30 to the dynamic growth of the functional food market” please replace “these” with names of those factors.

In the next lines you have given “The plant-based food market is 31 expected to grow at a cumulative annual growth rate (CAGR) of 11.9% from 2020 to 2027 32 to reach $74.2 billion by 2027” please mention about the country or is it for worldwide?

In next paragraph starting with “Probiotic products are one of the examples of functional food” please mention some names of probiotic products?

Please divide this sentence in subsentences as it is too long “The application of fermentation tech- 39 nology to plant – origin food matrix processing and the development of novel probiotic 40 fermented products not only increase the added value of fruit and vegetables, but also 41 organically combine probiotics and their active metabolites with prebiotics (dietary fiber, 42 etc.), thereby promoting intestinal health as well as preventing and relieving chronic 43 diseases. It is also indicating an important development direction for the probiotics in- 44 dustry in the future” I think which is grammatically not suitable.

Again in next paragraph same problem, Authors are requested to please thoroughly check the whole article and remove its grammatical mistakes.

Please mention about bacterial strains? Have you isolated  it or collected in isolated form? Secondly if isolated then which method you have used?

Results need some major revision, please compare your results and give the importance and reasoning in each parameters.

I think its better to explain separate discussion for each parameters along with its comparison and importance.

Please recheck thoroughly the whole article and improve its grammatical mistakes.

Recheck references according to the journal guidelines.

Author Response

(The authors gave the same response as above.)
